# Identifying Military Service Status in Electronic Healthcare Records from Psychiatric Secondary Healthcare Services: A Validation Exercise Using the Military Service Identification Tool

**DOI:** 10.3390/healthcare11040524

**Published:** 2023-02-10

**Authors:** Daniel Leightley, Laura Palmer, Charlotte Williamson, Ray Leal, Dave Chandran, Dominic Murphy, Nicola T. Fear, Sharon A. M. Stevelink

**Affiliations:** 1King’s Centre for Military Health Research, King’s College London, Weston Education Centre, Cutcombe Road, London SE5 9RJ, UK; 2Biomedical Research Centre (BRC), Institute of Psychiatry, Psychology and Neuroscience, King’s College London, London SE58AB, UK; 3Combat Stress, Tyrwhitt House, Oaklawn Road, Leatherhead, London KT22 0BX, UK; 4Academic Department of Military Mental Health, King’s College London, Weston Education Centre, Cutcombe Road, London SE5 9RJ, UK; 5Department of Psychological Medicine, Institute of Psychiatry, Psychology and Neuroscience, King’s College London, London SE58AB, UK

**Keywords:** electronic health records, mental health, armed forces, secondary mental healthcare, national health service, United Kingdom, veterans, military service identification tool

## Abstract

Electronic healthcare records (EHRs) are a rich source of information with a range of uses in secondary research. In the United Kingdom, there is no pan-national or nationally accepted marker indicating veteran status across all healthcare services. This presents significant obstacles to determining the healthcare needs of veterans using EHRs. To address this issue, we developed the Military Service Identification Tool (MSIT), using an iterative two-staged approach. In the first stage, a Structured Query Language approach was developed to identify veterans using a keyword rule-based approach. This informed the second stage, which was the development of the MSIT using machine learning, which, when tested, obtained an accuracy of 0.97, a positive predictive value of 0.90, a sensitivity of 0.91, and a negative predictive value of 0.98. To further validate the performance of the MSIT, the present study sought to verify the accuracy of the EHRs that trained the MSIT models. To achieve this, we surveyed 902 patients of a local specialist mental healthcare service, with 146 (16.2%) being asked if they had or had not served in the Armed Forces. In total 112 (76.7%) reported that they had not served, and 34 (23.3%) reported that they had served in the Armed Forces (accuracy: 0.84, sensitivity: 0.82, specificity: 0.91). The MSIT has the potential to be used for identifying veterans in the UK from free-text clinical documents and future use should be explored.

## 1. Introduction

The United Kingdom’s (UK) veteran population, defined by the British Government as those who have served in the military for at least one day [1], is estimated to be 2.5 million [2]. According to the 2021 census of England and Wales, 1.9 million people reported a veteran status [3]. UK veterans receive their healthcare from the National Health Service (NHS) alongside civilian counterparts, with their care recorded in local, regional, and national Electronic Healthcare Records (EHRs) [4]. EHRs—structured and unstructured (i.e., free-text, self-reported questionnaires)—can be used to evaluate disease prevalence, perform epidemiological analyses, investigate the quality of care, improve clinical decision-making, and employed for other research purposes [5,6,7,8]. The use of EHRs can also overcome challenges related to the recruitment and retention of veterans in research [9].

Whilst most veterans transition out of the military without experiencing any difficulties, a sizeable minority report a range of mental health problems [10,11,12]. Some of these are a result of their military service [13]. Research has shown that 93% of personnel who report having mental health difficulties seek some form of help for their problems, with the majority seeking informal support [14,15]. Currently, there is no universal pan-national or nationally accepted marker in UK EHRs to identify veterans, neither is there a requirement for healthcare providers to record veteran status or ask individuals about their potential military employment. This lack of data prohibits an evaluation of the healthcare needs of those who have served in the Armed Forces [16].

The ability to identify those who serve in the military would allow a comparison between those with and without military experience in relation to their demographic characteristics, physical and mental health outcomes, and healthcare utilization. Prior research has similarly conducted comparisons between the general population and specific cohorts, such as those with chronic mental illness [17,18], and emergency service workers [19]. The ability to identify military veterans could further enable primary and secondary healthcare services to target known areas of concern, such as alcohol use, with novel interventions [11,15]. Only three studies have analyzed secondary care in the UK using military samples [4,20,21]. These studies identified veterans via external sources, manual human review, and the use of data linkage algorithms. The development of the MSIT [22], a Natural Language Processing (NLP) tool, enabled the use of a machine learning tool to analyze free-text clinical documents. 

Veterans were manually identified using the South London and Maudsley (SLaM) Biomedical Research Centre (BRC) Clinical Record Interactive Search (CRIS) database holding secondary mental healthcare electronic records for the SLaM NHS Foundation Trust. An iterative approach was then followed. First, a structured query language (SQL) method was developed, which was refined using NLP and machine learning to create the MSIT, a tool designed to identify if a patient was a non-veteran or veteran [20,22]. We obtained an accuracy of 0.93 in correctly predicting non-veterans and veterans, a positive predictive value of 0.81 and a sensitivity of 0.75 using this approach. This method informed the second stage, which was the creation of the MSIT using machine learning, which, when tested, obtained an accuracy of 0.97, a positive predictive value of 0.90 and a sensitivity of 0.91.

Given the large-scale use of free text in patient record systems and the vital role they play in clinical decision-making and describing patient characteristics, it is important this recording is accurate. Variations in the methods of notetaking among different healthcare professionals—combined with local, regional and national differences—have created data quality issues that could impact the use of automated text analysis tools [23,24]. The objective of this study was to therefore validate the accuracy of free-text medical documents used to assess the performance of the MSIT by contacting participants to verify the accuracy of their military service status. It is important to note that the MSIT tool identifies military service. Since serving personnel transition from military to civilian healthcare services once leaving the military, almost all positive identifications of military service made by the MSIT will relate to veterans.

## 2. Background

Routinely collected EHRs can be used to evaluate disease prevalence, monitor disease spread, facilitate epidemiological analyses [25], and improve clinical decision-making which can influence patient outcomes [5,6]. EHRs function as a single integrated standardized longitudinal electronic version of the traditional paper health record and are held by hospitals, clinics, and other healthcare providers across the UK [4,16]. EHRs combine structured fields such as test results, or questionnaire responses with unstructured fields which contain free-text medical notes or communications (e.g., letters sent to patients).

In recent years, there has been a growth in the use of EHRs in the field of health data analytics either via NLP, or machine learning [26]. The use of these innovative approaches follows two main themes: to generate knowledge to improve the effectiveness of treatment, or to predict the outcome of treatment and diagnoses, or a combination of both [27].

In the context of NLP algorithms, the two main themes specified are used to perform syntactic representation (e.g., tokenization, sentence, and structure detection), extract specific information of themes (e.g., identify depressive symptoms or represent text in a structured form [28]), capture meaning from documents (e.g., sentiment of statements or free-text) and detect relationships (e.g., between diseases and conditions [29]).

For example, Downs et al. (2017; [30]) used NLP to detect keywords associated with suicidal ideation or attempted suicide in adolescents with autism spectrum disorders with a high degree of accuracy. A similar approach was taken by Al-Harras et al. (2021; [31]) where NLP was used to establish the presence of motor signs (e.g., gait, rigidity, tremor) using free-text clinical notes in patients with a dementia diagnosis. To identify motor signs, the authors developed an annotation pipeline where a set of documents were manually labeled, keywords extracted, and used as a gold standard reference when analyzing future documents. By identifying motor signs, the authors were able to identify the co-morbidity profiles of patients experiencing these symptoms and ascertained associations with survival and hospitalization. Irving et al. (2021; [32]) used NLP to identify gender differences in clinical presentations and illicit substance use. The authors developed an NLP framework that used a labeled corpus to extract information from free-text unstructured medical records to create a structured dataset that could then be used for analysis. Using this approach, they were able to identify clear differences between genders in clinical presentation, and in the substances used.

Approaches in recent years have also used machine learning combined with NLP to analyze large datasets. Kapadi et al. (2022; [33]) developed a machine learning and NLP framework to predict the inpatient risk of re-admission based on free-text clinical notes. The authors devised a framework where clinical notes were analyzed and coded using NLP, where term frequency-inverse document frequency vectors were computed and trained a Random Forest machine learning model. While the framework did not yield highly accurate results, it was able to analyze free-text clinical notes at scale. Conversely, Han et al. (2022; [34]) developed a framework where they were able to automatically detect social determinants of health using NLP and machine learning. First, they manually annotated over 3000 free-text notes and then trained a range of complex machine-learning models. With this framework, they were able to produce Area Under the Curve results of 0.97.

EHRs are a powerful resource when combined with NLP and machine learning; this approach may enable health practitioners, decision-makers and researchers to monitor and improve admissions, clinical decision-making patient outcomes, and quality of care. These algorithms also have the potential to identify current and former occupations, including military service, thus enabling a suite of research that could examine the health characteristics of different occupational groups.

## 3. Materials and Methods

### 3.1. Data Source—Clinical Record Interactive Search (CRIS) System

The Clinical Record Interactive Search (CRIS) system provides de-identified EHRs from the SLaM NHS Foundation Trust, a secondary and tertiary mental healthcare provider serving a geographical catchment of approximately 1.3 million residents across 4 south London boroughs (namely Lambeth, Southwark, Lewisham, and Croydon) [25].

### 3.2. Military Service Identification Tool (MSIT)

A machine learning classification framework underpins the MSIT and is responsible for making predictions. It was developed in Python using the Natural Language Processing Toolkit (version 3.2.5) [35] and Scikit-learn (version 0.20.3) [36]. A demonstration of MSIT is available via GitHub (https://github.com/DrDanL/kcmhr-msit, accessed on 8 February 2023). For this article, we provide a brief overview of the MSIT.

A gold standard manually labeled dataset was created to train the MSIT. This labeled dataset included terms such as “veteran”, “army”, and “served in the forces”. Once documents had been identified, free-text documents were preprocessed to remove:(1)punctuations (using regular expressions);(2)words/phrases related to another individual’s military service;(3)stop words and frequently occurring terms (except military terms); and(4)word/phrases that may cause confusion with correctly identifying a veteran.

The remaining features were then converted into term frequency-inverse document frequency features. The classification framework was trained to identify veterans based on the use of military terms and phrases with the outcome being binary (1: veteran and 0: non-veteran; see for more information [22]).

To improve the true positive rate of the MSIT and reduce the potential for false positives, a postprocessing of the outcome was applied based on a set of rules. For each document that was classified as containing indicators of military service, an SQL operation was performed to ensure the document used a military term or phrase.

### 3.3. Military Service Identification Tool (MSIT) Validation Approach

Whilst the MSIT has high precision, it is important to ensure that the MSIT can identify “true” military veterans and that the underlying data used to train the MSIT are accurate. This was achieved by sending an online survey to a sample of patients in the SLaM NHS Foundation Trust to determine their self-reported veteran/non-veteran status and to compare this to their MSIT classifications. Responses allowed us to validate the MSIT algorithm’s predictive performance. Incorrect classifications would provide further information with which to refine the algorithm and improve its accuracy. To undertake this evaluation, the following steps were followed:**Sample:** The MSIT was run over a batch of 779,944 records from patients who had used SLaM NHS Foundation Trust services since January 2017. This month marked the inception of the Consent for Consent (C4C) mechanism that records patients who consent to be contacted for research purposes. From these records, the BRC identified patients who were listed (1) as “Alive”; (2) aged 18 years or older; (3) had given C4C; (4) did not have indicators of dementia or psychosis according to their diagnostic codes; (5) had an email address or mobile telephone number; (6) were able to communicate in English without an interpreter; and (7) if they were active patients, were approved by their care coordinator to take part in the study.**MSIT Execution:** Once the sample had been identified, all free-text clinical documents relating to their care in SLaM were analyzed using the MSIT. Each patient was evaluated as being a military veteran or non-veteran based on their medical notes.**Recruitment:** Recruitment spanned February to June 2022. The research team performed another manual screening to detect any changes or further details regarding patients’ eligibility criteria (e.g., becoming an inpatient, now deceased, changes to their C4C status, and/or apparent communication needs that may prevent them from being able to take part in the survey). Participants were contacted in the first instance by email and, if this was not available, by text. Reminder emails and texts were sent and if participants requested not to be contacted, the researchers updated their C4C status.**Data collection**: Patients were invited to take part in the study via email or text, which included the survey link. After consenting, participants were asked a single question: “Have you ever served in the Armed Forces (military)?” to which they could reply “Yes” or “No”. If participants endorsed having served in the Armed Forces, they were asked follow-up questions about their military characteristics to further develop the MSIT, and to provide context for understanding any inaccurate misclassifications. These questions collected branch of service, rank, length of service, and regular/reserve status. Data relating to patients’ EHRs accessible via SLaM Electronic Patient Journey System (ePJS) were kept separately from their survey responses.**Prize draw:** Participants who completed the survey were entered into a prize draw for 26 e-vouchers comprised of 20 × £10, 5 × £20, and 1 × £50.

### 3.4. Statistical Analysis

All analyses were performed using STATA 16.1 MP (StataCorp, College Station, TX, USA). The positive predictive value was defined as the proportion of correctly identified true veterans over the total number of true veterans identified by the MSIT. Sensitivity was defined as the proportion of non-veterans identified by the MSIT over the total number of actual non-veterans (identified by patient report); specificity was determined as the proportion of veterans identified by the MSIT over the total number of actual veterans. Accuracy was measured using the Youden Index [37], which considers sensitivity and specificity ([38]; summation minus 1), which results in a value that lies between 0 (absence of accuracy) and 1 (perfect accuracy).

### 3.5. Ethical Approval

Ethical approval was given by the East of Scotland Research Ethics Service within the NHS Research Ethics Service (Ref: 20/ES/0060). Approvals were obtained by the SLaM Research and Development Office at King’s College London (Ref: R&D2020/029).

## 4. Results

### 4.1. Sample and MSIT Execution

Each stage of sampling, recruitment and inclusion/exclusion is depicted in Figure 1. Between March 2022 and June 2022, the MSIT was run over 779,944 free-text EHRs held by the BRC CRIS system, representing 141,762 patients. Each document was assigned a flag (1 = veteran, 2 = non-veteran), and this was aggregated at patient level so that any presence of a veteran flag in a document meant the patient was identified as being a veteran.

After applying the automated exclusion criteria listed in Section 3.3, 1684 (1.2%) remained for manual confirmation of their eligibility. After this process, 902 (53.6%) patients were invited to take part in the study with 149 (16.5%) providing a response to the questionnaire. Three of these responses had to be excluded as they were duplicate entries.

### 4.2. Validation Results

In total, 146 participants provided eligible responses to the questionnaire (see Table 1). Of these, 112 (76.7%) were classified by the MSIT as non-veterans and 34 (23.3%) were classified as veterans. When corroborating survey responses and MSIT classifications, we found 84.2% of the sample was accurately categorized by the MSIT (*n* = 122/146). A sensitivity and specificity analysis were performed to determine how many veterans and non-veterans were misclassified. Overall, 23 true non-veterans were inaccurately categorized by the MSIT as veterans, and 1 veteran was inaccurately categorized by the MSIT as non-veteran. We found:The sensitivity of the MSIT was 0.83.The specificity of the MSIT was 0.92.

A manual investigation of inaccurate classifications found that the MSIT had a high degree of accuracy with some exceptions. In minor examples of misclassification, the MSIT may be prone to assigning non-veterans as veterans. The most common reasons for misclassification included the mentioning of military family members and support received by the Salvation Army.

**Table 1 healthcare-11-00524-t001:** MSIT classifications compared with patient-reported classification.

Outcome	True Non-Veteran	True Veteran	Total
MSIT Non-veteran	111	1	112
MSIT Veteran	23	11	34
Total	134	12	146

## 5. Discussion

This validation study has demonstrated that it is possible to identify veterans from free-text clinical notes using the MSIT. The MSIT tool performed well, as indicated by its high sensitivity and specificity. To the authors’ knowledge, this is the only study to have developed, applied, tested, and validated an NLP and machine learning framework for the identification of veterans in the UK using a large psychiatric database. Notably, by assessing the validity of the underlying free-text record, we have also demonstrated accuracy in the identification of records belonging to veterans.

In the absence of a universal marker denoting prior military service, there has been no systematic way to identify and to thus examine veteran populations in mental healthcare services. The current Veterans’ Strategy Action Plan reinforces the importance of better understanding the health and well-being needs of the veteran community [39]. The ability to identify veterans may advance investigations into the mental health characteristics of veterans accessing these services, and their navigation through, and use of, such services. This knowledge is vital for delivering commitments to the veteran community as outlined in the NHS Long Term Plan, such as inclusive access to services, improving provision, and identifying and addressing potential health disparities [40]. Asking a simple question—such as “have you ever served in the UK Armed Forces?”—and have the response recorded could yield significant public health benefits. While this question is asked at a local level by some General Practitioners, it is not recorded routinely in secondary care.

EHR-based Case Registers, such as CRIS, function as single, complete, and integrated electronic versions of traditional paper health records [4]. These registers have been positioned as a ‘new generation’ for health research and, since the year 2000, are mandatory across the UK [4]. The methodological advantages of Case Registers—including their longitudinal nature, largely structured fields, and detailed coverage of defined populations—make them an ideal research and monitoring tool [41]. There is also the potential to have these Case Registers linked to multiple healthcare providers, third-sector charitable organizations, and medication dispensing providers to further improve the overall support and clinical delivery for patients.

EHRs in mental healthcare provide extremely rich material and analyses of their data can reveal patterns in healthcare provisions, patient profiles and mental and physical health problems [4,42]. This is hugely advantageous for investigating vulnerable sub-groups within the wider population [25,43,44].

### 5.1. Strengths and Limitations

This study has demonstrated that it is possible to identify veterans who accessed secondary mental healthcare services in the UK by using a Case Register. The MSIT was able to identify potential veterans with high precision, with 84.2% of cases correctly identified when corroborated against patients’ self-reported status. A key strength of the MSIT was the exploitation of NLP and the annotation of a large corpus of medical records. This is advantageous for automating the process of identifying veterans, as well as reducing the possibility of human error, and conscious/unconscious biases, and overcoming challenges when using military cohorts linked to Case Registers [4,45].

At present, the MSIT is the only tool implemented by CRIS using C4C to validate the integrity of the patient record and validate reported status [46]. The methodology reported in this manuscript could aid, and further support, future studies using C4C to validate NLP tools and patient record validity.

The MSIT does not rely on any coding structure or predefined fields and solely uses free-text documents, which broadens its potential applications to areas like diagnoses, occupations, and identifying ethnicity. This may allow studies to assess the specific characteristics of veterans in the SLaM sample compared to their non-veteran counterparts, as undertaken by Mark et al. (2019; [20]).

This current study, however, highlights some limitations of self-report data, whether this is inputted by healthcare professionals or reported by the patient themselves. For instance, veterans may be misclassified as non-veterans if they did not disclose their veteran status during consultations, if this was not recorded, or if it was misreported by clinicians. Whilst notes on occupational history are commonly taken during consultations, any reports of military service during consultations are also not verified using military records, and therefore there may be false disclosures.

In terms of the MSIT’s performance, there some evidence of non-veterans being misclassified as veterans when keywords referred to other contexts, e.g., military family members, military metaphors, or the Salvation Army. Minor revisions to the keywords used by the tool are required before future use, however, the current findings suggest that the MSIT does not require any substantial changes.

Despite maximizing the sampling pool to improve our chances of finding eligible veterans, our attempts did not yield a large veteran sample (*n* = 12 “true” veterans comparative to *n* = 112 “true” non-veterans). This was not deemed a substantial problem since the research team deliberately sought to oversample non-veterans. This was decided because erroneous classifications of non-veterans would contaminate a “true” veteran sample for future analyses. In addition, the smaller sample size of veterans reflects the reality that non-veterans outnumber veterans in the general population.

This study was not able to establish whether veterans or non-veterans were disproportionately impacted by specific exclusion criteria of this study (e.g., being less likely to have C4C or being a current inpatient). This may prohibit specific individuals, e.g., those with more complex mental health issues who may not consent to participate in research, from being represented in future analyses. This is an important consideration for future research.

### 5.2. Implications

To address the inconsistencies in whether veteran status is recorded, a military service marker in the case registers and similar NHS databases could be implemented. Although time-consuming, it is possible that this could be verified by referencing UK Ministry of Defence records. This could be accompanied by broader educational efforts to help clinicians better support veterans in their care, to have a more in-depth understanding of the unique mental health needs of this population, and to be aware of the benefits of recording veteran status for referring into bespoke and specialist services. Any endeavor to ascertain veteran status must acknowledge that this can be a highly sensitive and personal attribute, especially in areas such as Northern Ireland [47].

Additionally, the MSIT could be customized and further tested to identify other occupational groups, including those that are similarly exposed to potentially traumatic events, such as first responders; who have similar work-related patterns, such as oil rig workers, or who experience specific mental health issues, such as suicide among construction workers [48].

Whilst the MSIT was successful in the context of the SLaM NHS Foundation Trust, further work is required to refine the tool to function on other datasets from other NHS Trusts. To that end, we have released the source-code of the tool. As with any other effort to access and test tools on EHRs, the researchers encountered many bureaucratic and governance hurdles, making the research an extensive and lengthy endeavor. This is noteworthy for future research seeking to test, develop and utilize the MSIT and other applications. The potential benefits of the MSIT, however, remain. Principally, the MSIT allows for the identification of veterans until a mandatory field is introduced, and for its continued use in retrospective analyses.

## 6. Conclusions

We have shown that it is possible to identify veterans using NLP in electronic healthcare records of a secondary mental health service. The MSIT was able to identify military veterans with high accuracy and precision. This work has demonstrated that the MSIT can be used in a healthcare setting and was successful in exploiting 800,000+ free-text medical records. With the ability to identify military veterans, we can now conduct analyses comparing the health and well-being needs of this important cohort with the public, and other high-intensity occupations. With further refinement, the MSIT can be implemented in other electronic healthcare systems and to possibly identify other occupational groups.

## Figures and Tables

**Figure 1 healthcare-11-00524-f001:**
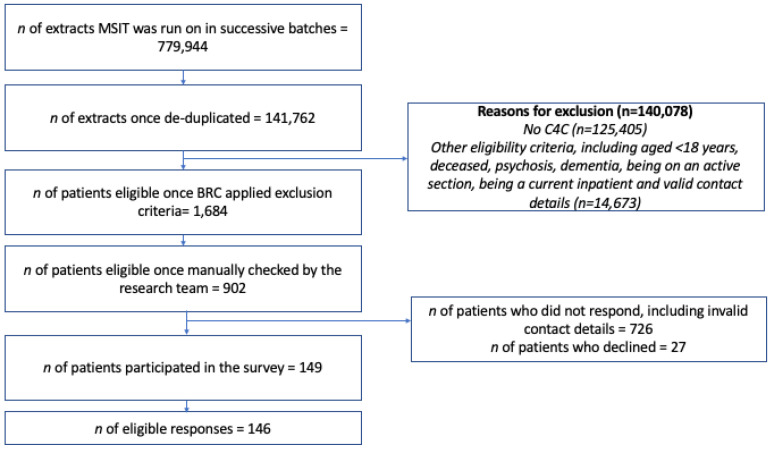
Flow diagram of the recruitment process for the MSIT validation study.

## Data Availability

The datasets used in this study are based on patient data, which are not publicly available. Although the data are pseudonymized, that is, personal details of the patient are removed, the data still contain information that could be used to identify a patient. Access to these data requires a formal application to the CRIS Patient Data Oversight Committee of the NIHR Biomedical Research Centre. On request and after suitable arrangements are put in place, the data and modeling employed in this study can be viewed within the secure system firewall. The corresponding author can provide more information about the process.

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
