# Peer review of "Identifying Military Service Status in Electronic Healthcare Records from Psychiatric Secondary Healthcare Services: A Validation Exercise Using the Military Service Identification Tool"

_healthcare, 2023, doi:10.3390/healthcare11040524_

Round 1

Reviewer 1 Report

General

This paper is valuable in that it demonstrates a technique or tool that can be used, with modification, to potentially retrospectively identify many factors in a psychiatric patients history which may prove valuable in epidemiological studies. It would also specifically help identify outcomes for the veteran community on retrospective analysis of notes. This would include differences in diagnosis or 'sectioning'.

I was not clear if the term 'veteran' was itself used in the MSIT, and if so whether running that single word search alone would have made a difference or excluding that term from the search would have made a difference? Clearly those entering the electronic data in the health record may have identified the patent as a veteran and appropriately recorded it which in itself would then cause a cascade of other terms identified by the search. A comment on this would be appreciated. Without this additional comment a significant bias is introduced.

Of course the ultimate comment which should be included is a recommendation that all health records, held nationally, and able to be interrogated should be introduced recording that a patient has been asked 'have they ever served' as part of routine admission in collecting other demographic details. This would obviate the need for this technique in future.

Major Comments

The title should state this is in Psychiatric Secondary Care Electronic Mental Health Care records - psychiatric health care records are significantly different from other secondary health care services in terms of depth, occupational, social etc... histories. It is unlikely this would work in other health care settings (eg orthopaedics) and should not be implied.

Lines 85 - 86 . The statement that all service personnel in secondary mental health are discharged is not accurate, this is not true and should be amended.

Minor

Line 19 no 'pan-national' or 'nationally accepted marker covering all health care services' - as there is a national primary care marker.

Line 54 the word 'help' should end the sentence.

Author Response

Comment: This paper is valuable in that it demonstrates a technique or tool that can be used, with modification, to potentially retrospectively identify many factors in a psychiatric patients history which may prove valuable in epidemiological studies. It would also specifically help identify outcomes for the veteran community on retrospective analysis of notes. This would include differences in diagnosis or 'sectioning'.

Response: We thank the Reviewer for providing helpful and supportive comments to our manuscript.

Comment: I was not clear if the term 'veteran' was itself used in the MSIT, and if so whether running that single word search alone would have made a difference or excluding that term from the search would have made a difference? Clearly those entering the electronic data in the health record may have identified the patent as a veteran and appropriately recorded it which in itself would then cause a cascade of other terms identified by the search. A comment on this would be appreciated. Without this additional comment a significant bias is introduced.

Response: We have modified the manuscript to include further information on how MSIT operated. The term veteran (and similar terms) was included as part of MSIT. In the development of the MSIT we consulted with our colleagues, who are also practitioners working within the South London and Maudsley Trust (SLaM), as to how EHRs are used; it appeared the word ‘veteran’ is infrequently use and instead it is more common to describe someone has having “served in the Army”, “they were discharged from the RAF”. To reflect the types of keywords used, we have made the following inclusion:

“A gold standard manually labelled dataset was created to train the MSIT. This la-belled dataset included terms such as “veteran”, “army”, and “served in the forces”.”

These changes can be viewed on page 4, between line 151 and 171, where we have further clarified how MSIT was developed.

Comment: Of course the ultimate comment which should be included is a recommendation that all health records, held nationally, and able to be interrogated should be introduced recording that a patient has been asked 'have they ever served' as part of routine admission in collecting other demographic details. This would obviate the need for this technique in future.

Response: We have amended the manuscript to specifically reference the utility of asking patients if they have ever served in the armed forces. We have included the following statements:

Page 2, between line 54 and 88:

“Currently, there is no universal pan-national or nationally accepted marker in UK EHRs to identify veterans, nor is there a requirement for healthcare providers to record it or ask individuals about their potential military employment. This lack of data prohibits an evaluation of the healthcare needs of those who have served in the Armed Forces [16].”

Page 7, between line 290 and 293:

“The ability to ask even a simple question – such as “have you ever served in the UK Armed Forces” – and have the response recorded could yield significant public health benefit. While this question is asked at a local level by some General Practitioners, it is not recorded routinely in secondary care.”

Comment: The title should state this is in Psychiatric Secondary Care Electronic Mental Health Care records - psychiatric health care records are significantly different from other secondary health care services in terms of depth, occupational, social etc... histories. It is unlikely this would work in other health care settings (eg orthopaedics) and should not be implied.

Response: We have modified the title based on the suggestions of the Reviewer. The new title is as follows:

“Identifying military service status in electronic healthcare records from psychiatric secondary healthcare services: A validation exercise using the Military Service Identification Tool”

Comment: Lines 85 - 86 . The statement that all service personnel in secondary mental health are discharged is not accurate, this is not true and should be amended.

Response: We have modified the manuscript to include the following statement on Page 2, between line 90 and 93:

It is important to note that the MSIT tool identifies military service. Since serving personnel transition from military to civilian healthcare services once leaving the military, almost all positive identifications of military service made by the MSIT will relate to veterans.”

Comment: Line 19 no 'pan-national' or 'nationally accepted marker covering all health care services' - as there is a national primary care marker.

Response: We have amended this statement to include suggested revisions provided by the Reviewer. These changes have been reflected on Page 1: line 19 and Page 2: line 54.

Line 54 the word 'help' should end the sentence.

Response: We have included this word. This change has been made on Page 2, line 56.

Reviewer 2 Report

1) The authors may want to add background section with recent papers.

2) The authors need to extend the conclusion section. 

3) Please discuss future studies and the limitations of the study.

4) There are some grammatical or spelling errors in the manuscript. For instance: "2.4 Statstical analysis", please correct it and check all text carefully 

Author Response

Response: We thank the Reviewer for providing the comments below. We hope our interpretation of these comments, and our responses, adequately address the reviewer's concerns

Comment: The authors may want to add background section with recent papers.

Response: We have added a “Background” section into the manuscript detailing key NLP papers which highlight its use and utility in the field of healthcare. This has been added on Page 2, between line 95 and 143.  

Comment: The authors need to extend the conclusion section. 

Response: We have modified the manuscript to expand upon our concluding remarks. This change has been made on Page 8, between line 375 and 383. For ease of reference, this excerpt is cited below:

“We have shown that it is possible to identify veterans using NLP in electronic healthcare records of a secondary mental health service. The MSIT was able to identify military veterans with high accuracy and precision. This work has demonstrated that the MSIT can be used in a healthcare setting and was successful in exploiting 800,000+ free-text medical records. With the ability to identify military veterans, we now can conduct analyses comparing the health and wellbeing needs of this important cohort with the public, and other high intensity occupations. With further refinement, the MSIT can be implemented in other electronic healthcare systems and to possibly detect other occupational groups.“

Comment: Please discuss future studies and the limitations of the study.

Response: In this manuscript, we had included a strengths and limitations section which outlined limitations of the study, and the potential for future studies. In response to this advice, we have made several changes to this section to improve clarity.  Please see Page 7 line 307 to Page 8 line 349. 

Comment: There are some grammatical or spelling errors in the manuscript. For instance: "2.4 Statstical analysis", please correct it and check all text carefully 

Response: We have revised the text throughout to correct grammatical and spelling errors. 

Reviewer 3 Report

My comments:
1. The topic of this paper is interesting and innovate and it will contribute in related research field.

2. A section of “Related Works” or “Literature Review” is necessary for this paper.

3. The “2. Materials and Methods” and “3. Results” are too brief and should be more detailed.

4. The “5. Conclusions” must be reinforced more. For example, the contributions to academic research as well as theoretical implications and research limitations.

Author Response

We thank the Reviewer for providing the comments below. We have addressed Reviewer 3’s feedback in conjunction with feedback from Reviewers 1 and 2.

Comment: The topic of this paper is interesting and innovate and it will contribute in related research field.

Response: We thank the Reviewer for providing this encouragement. We hope our modifications to the manuscript have further strengthened our work.

Comment: A section of “Related Works” or “Literature Review” is necessary for this paper.

Response: We have added a “Background” section into the manuscript detailing key NLP papers which highlight its use and utility. This has been added on Page 2, between line 95 and 143. 

Comment: The “2. Materials and Methods” and “3. Results” are too brief and should be more detailed.

Response: We have extended the manuscript to provide further clarity and context to our work. This includes multiple changes to each section which has resulted in our manuscript increasing in length by approximately 1000 words.    

Comment: The “5. Conclusions” must be reinforced more. For example, the contributions to academic research as well as theoretical implications and research limitations.

Response: We have modified the manuscript to expand upon our concluding remarks. This change has been made on Page 8, between line 375 and 383. For ease of reference, this excerpt is cited below:

“We have shown that it is possible to identify veterans using NLP in electronic healthcare records of a secondary mental health service. The MSIT was able to identify military veterans with high accuracy and precision. This work has demonstrated that the MSIT can be used in a healthcare setting and was successful in exploiting 800,000+ free-text medical records. With the ability to identify military veterans, we now can conduct analyses comparing the health and wellbeing needs of this important cohort with the public, and other high intensity occupations. With further refinement, the MSIT can be implemented in other electronic healthcare systems and to possibly detect other occupational groups.“

Round 2

Reviewer 3 Report

This paper is qualified to be published.